# Clinical Predictors and Prognostic Significance of Pathologic Disease Upstaging at Radical Cystectomy in Patients with Muscle-Invasive Bladder Cancer

**DOI:** 10.3390/cancers17193265

**Published:** 2025-10-09

**Authors:** Salvador Jaime-Casas, Wesley Yip, Daniel J. Lama, Vitor Goes, Miguel Zugman, Koral Shah, Regina Barragan-Carrillo, Hedyeh Ebrahimi, Daniela V. Castro, Yu Jun Li, Benjamin Mercier, JoAnn Hsu, Xiaochen Li, Clayton S. Lau, Kevin G. Chan, Bertram E. Yuh, Alexander Chehrazi-Raffle, Sumanta K. Pal, Abhishek Tripathi

**Affiliations:** 1Department of Medical Oncology & Therapeutics Research, City of Hope Comprehensive Cancer Center, 1500 E. Duarte Road, Duarte, CA 91010, USA; sjaimecasas@coh.org (S.J.-C.); dacastro@coh.org (D.V.C.); achehraziraffle@coh.org (A.C.-R.);; 2Division of Urology and Urologic Oncology, Department of Surgery, City of Hope Comprehensive Cancer Center, Duarte, CA 91010, USA

**Keywords:** bladder cancer, upstaging, outcomes, neoadjuvant, cystectomy, muscle-invasive

## Abstract

**Simple Summary:**

Clinical staging in patients with bladder cancer can have inherent limitations, often understaging disease burden. In such cases, surgical pathology confirms the disease grade and can often reveal upstaging of tumor invasion. We decided to evaluate the clinical and pathological predictors of pathologic upstaging (pUS) in a cohort of patients who underwent robot-assisted radical cystectomy (RARC), stratifying groups based on neoadjuvant chemotherapy receipt status. Our results revealed that preoperative hydronephrosis was the strongest predictor of pUS, independent of other baseline covariates, highlighting the need for improved preoperative risk stratification strategies for patients with MIBC undergoing RARC.

**Abstract:**

Introduction: Staging inaccuracies in muscle-invasive bladder cancer (MIBC) can lead to undertreatment or overtreatment. We evaluated clinical and pathological predictors of pathologic upstaging (pUS) stratifying by neoadjuvant chemotherapy (NAC) receipt among patients undergoing robot-assisted radical cystectomy (RARC). Methods: We included patients with MIBC (≥cT2N0M0) who underwent RARC from February 2004 through October 2020. Patients were grouped as (1) pUS with NAC, (2) pUS without NAC, and (3) no pUS (reference). Baseline characteristics were summarized using descriptive statistics. Logistic regression assessed the association between baseline characteristics and odds for upstaging. Kaplan–Meier method estimated overall survival (OS) and recurrence-free survival (RFS), and log-rank test compared the survival distribution between groups. Univariable and multivariable Cox regression models identified variables associated with OS and RFS. Results: Among 277 patients, 38.6% (n = 107) were upstaged with NAC (n = 37) or without NAC (n = 70). Most were male (79%), white (72%), and had cT2 stage (85%). Median age at surgery was 72 yrs. Preoperative hydronephrosis showed higher odds of upstaging [OR 2.24 (95% CI, 1.31–3.81), *p* = 0.003]. pUS with NAC [HR 1.99 (95% CI, 1.23–3.22), *p* = 0.005] and without NAC [HR 3.18 (95% CI, 2.21–4.55), *p* < 0.001] predicted worse OS (33.5 vs. 18.8 mos) compared to patients without pUS (135.3 mos). pUS with NAC [HR 2.49 (95% CI, 1.58–3.94) *p* < 0.001] and without NAC [HR 3.02 (95% CI 2.11–4.31), *p* < 0.001] predicted worse RFS. Conclusions: Preoperative hydronephrosis was the strongest predictor for pUS, independent of other baseline covariates. This highlights the need for better pre-operative risk stratification strategies for patients with MIBC undergoing RARC.

## 1. Introduction

Bladder cancer is one of the most aggressive genitourinary malignancies. It represents the ninth most common cancer worldwide, with 75% of cases accounting for non-muscle invasive bladder cancer and 25% representing muscle-invasive bladder cancer (MIBC) [1,2,3]. Given the aggressive natural history of MIBC, upfront systemic therapy and surgery are recommended to mitigate the increased risk of progression and cancer-specific mortality [4]. Cisplatin-based neoadjuvant chemotherapy (NAC) followed by radical cystectomy is the cornerstone of therapy as it significantly improves overall survival (OS) in patients with MIBC [5,6]. The SWOG-8710 trial established the benefits of this therapy by showing a higher rate of pathologic complete response and OS in patients receiving platinum-based chemotherapy compared to patients randomized to surgery alone [7]. More recently, the NIAGARA trial expanded the therapy armamentarium by showing improved event-free survival and OS with chemo-immunotherapy as opposed to chemotherapy alone in patients with MIBC [8,9,10]. In select patients, bladder-sparing approaches in the form of tri-modal therapy can be attempted. This incorporates maximal transurethral resection of bladder tumor (TURBT) with concurrent chemotherapy and radiation therapy. Although most studies have been retrospective, results have shown comparable outcomes with upfront radical cystectomy in select populations [11,12,13,14,15,16].

Despite recent advances in treatments incorporating multiagent regimens, the oncologic outcomes for this patient population remain suboptimal, with a 5-year OS rate of approximately 50–70% [11]. This can be partly explained by occult micro-metastatic disease at diagnosis as well as de novo resistance to existing NAC regimens. Consequently, initial clinical staging can be challenging for physicians, and incorporating risk-defining clinical and pathological variables is necessary to guide management and optimize patient survival. We investigate the prevalence of risk factors for pathologic disease upstaging and its association with oncologic outcomes in patients with MIBC treated with robotic-assisted radical cystectomy (RARC) with or without NAC.

## 2. Methods

### 2.1. Patient Population

We retrospectively reviewed an institutional review board-approved (IRB 05148) database of patients with MIBC treated with RARC from February 2004 through October 2020. We included patients with clinical (defined by cross-sectional computed tomography or cystoscopy/TURBT) T2-4N0M0 MIBC. Clinical staging was determined at the time of diagnosis by the treating urologist, based on cystoscopy/TURBT findings and cross-sectional imaging (CT or MRI), and was documented in the electronic medical record. Patients were excluded if they received non-cisplatin-based neoadjuvant chemotherapy, previous radiotherapy, trimodal therapy as part of a bladder-sparing approach, or had incomplete information. Patients with upper tract urothelial carcinoma (UTUC) were also excluded from this study. Hydronephrosis was considered tumor-related based on the evaluation of the treating physician, unless clearly attributable to another cause. Patients were further stratified based on the presence or absence of pathologic upstaging (defined as an increase in either T or N stage per surgical pathology report) at the time of RARC and whether they received NAC. Baseline characteristics were recorded for all patients. Age, body mass index (BMI), and preoperative renal function (creatinine, mg/dL) were continuous variables. Sex, race, American Society of Anesthesiology (ASA) score, NAC status, tumor histology, preoperative hydronephrosis, lymphovascular invasion (LVI) on TURBT, concomitant carcinoma in situ (CIS) on TURBT, and pathological node stage were considered categorical variables.

### 2.2. Statistical Analysis

Descriptive statistics were used to summarize baseline characteristics and clinicopathological data. Categorical data was reported as counts and percentages and continuous variables were reported as medians and interquartile (IQR) ranges. Differences in the distribution of baseline characteristics between the cohorts were assessed with the Chi-square/Fisher’s test for categorical variables and the analysis of variance (ANOVA) or Kruskal–Wallis test for continuous variables, depending on the results of a normality test. Patients without disease upstaging status (i.e., downstaged or no change difference between clinical and pathological stage) at the time of RARC served as the reference group. The primary endpoints included recurrence-free survival (RFS) and OS. RFS was defined as the time from surgery to local or distant disease recurrence based on histological or radiological evidence. OS was defined as the time from surgery to death due to any cause. Local and distant recurrence was defined as radiographic or histologic evidence of disease recurrence within or outside the pelvis, respectively. The Kaplan–Meier method was used to estimate OS and RFS, and the log-rank test was applied to compare the survival distribution between groups. Univariable and multivariable Cox regression models were used to identify variables associated with OS and RFS. Statistically significant variables found on univariable analysis were included in the multivariable model. Logistic regression models were used to assess the association between upstaging status and patients’ baseline clinical and pathologic characteristics to estimate the odds ratios (OR) for being upstaged after adjusting for covariates. All tests were two-sided, and statistical significance was considered as *p* < 0.05. All statistical analysis was performed with R statistical software (v.4.2.1).

## 3. Results

### 3.1. Patient Characteristics

A total of 277 patients with MIBC were included in the analysis, of which 45 (16%) received NAC, 107 patients (39%) were upstaged, and 8 patients (3%) were downstaged From the cohort of patients who experienced upstaging, 13.5% (n = 37) were upstaged and received NAC and 25.2% (n = 70) were upstaged without NAC. Among 45 patients who received NAC, 21 (47%) experienced upstaging. Among 118 patients who did not receive NAC (42%), 73 (62%) experienced upstaging. Overall, most patients were male (79%), white (72%), had pure-urothelial tumor histology (86%), and had cT2 disease (85%) at the time of diagnosis. Patients who were upstaged without NAC were older (median age: 77 years) compared to those who were upstaged and received NAC (median age: 69 years) and those who were not upstaged (median age: 71 years). Kruskal–Wallis test revealed a significant difference in age across the groups (*p* < 0.001). The proportions of patients with preoperative hydronephrosis differed significantly between groups, with a higher prevalence in upstaged patients who received NAC (43%) and those who did not (37%), compared to patients who were not upstaged (22%) (*p* = 0.008). Pathologic node stage differed significantly across the groups (*p* < 0.001), with a comparable frequency of pN1–3 in the upstaged groups (51% in upstaged and NAC, and 50% in upstaged and no NAC). The majority of patients who were not upstaged were node-negative on surgical pathology (pN0, 98%), whereas upstaged patients had substantially higher rates of nodal involvement (pN1–3), including 51% of those who received NAC and 50% of those who did not. We recorded a more pronounced numerical difference in pN1–3 rates between the upstaged and NAC (n = 19) and upstaged without NAC (n = 35) groups. Patients who were upstaged without prior NAC had higher preoperative creatinine levels (mg/dL) (1.17) compared to those upstaged with NAC (1.07) and those who were not upstaged (1.05, *p* = 0.02). We recorded no significant differences for concomitant CIS or LVI on TURBT. Patient baseline characteristics are shown in Table 1. Upstaging trends based on pathological node and tumor stage for the entire cohort are exemplified in Figure 1.

### 3.2. Survival Outcomes

Median follow-up time was 104 months (95% CI 88–124) for the not upstaged group, 109 months (95% CI 96-NE) for the upstaged and NAC groups, and 164 months (95% CI 112-NE) for the upstaged and no NAC group. Median OS was significantly longer for patients who were not upstaged (135.3 months; 95% CI, 99.8–183.0) compared to patients who were upstaged and received NAC (33.5 months; 95% CI, 21.1-NE) or who were upstaged without NAC (18.8 months; 95% CI, 13.5–32.2; *p* < 0.001). Similarly, median RFS was also significantly longer for patients who were not upstaged (108.6 months; 95% CI, 97.3–162.0), compared to patients who were upstaged and received NAC (21.1 months; 95% CI, 9.6–92.3) or patients who were upstaged without prior NAC (12.4 months; 95% CI, 10.4–18.1; *p* < 0.001). Kaplan–Meier curves for OS and RFS are shown in Figure 2.

### 3.3. Multivariable Analysis

Univariable and multivariable Cox regression models were adjusted for upstaging status, NAC completion status, age at surgery, ASA score, tumor histology, preoperative hydronephrosis, pathologic lymph node stage, concomitant CIS on TURBT, and LVI on TURBT. Upstaging without prior NAC was associated with significantly worse OS [HR 3.18; 95% CI, 2.21–4.55; *p* < 0.001], as was upstaging with prior NAC [HR 1.99; 95% CI, 1.23–3.22; *p* = 0.005]. Additional predictors of worse OS included older age at surgery [HR 1.03; 95% CI, 1.01–1.05; *p* = 0.004], presence of preoperative hydronephrosis [HR 1.44; 95% CI, 1.02–2.04; *p* = 0.03], and cT4 clinical stage [HR 3.18; 95% CI, 1.79–5.66; *p* < 0.001]. Predictors of worse RFS on multivariable analysis included upstaging without prior NAC [HR 3.02; 95% CI, 2.11–4.31; *p* < 0.001], upstaging with NAC [HR 2.49; 95% CI, 1.58–3.94; *p* < 0.001], older age at surgery [HR 1.03; 95% CI, 1.01–1.04; *p* = 0.003], preoperative hydronephrosis [HR 1.43; 95% CI, 1.02–2.01; *p* = 0.04], cT3 stage [HR 1.95; 95% CI, 1.18–3.23; *p* = 0.009], cT4 stage [HR 3.77; 95% CI, 2.19–6.47; *p* < 0.001], and elevated preoperative creatinine [HR 1.45; 95% CI, 1.05–2.00; *p* = 0.022]. Complete multivariable model results are presented in Table 2 and Table 3.

Logistic regression analysis demonstrated that patients with preoperative hydronephrosis had significantly higher odds of being upstaged than those without hydronephrosis [OR 2.24; 95% CI, 1.31–3.81; *p* = 0.003]. We recorded that every one-year increase in age was associated with a 2% increase in the odds of upstaging [OR 1.02; 95% CI, 1.00–1.05], although this finding did not reach statistical significance (*p* = 0.09). Logistic regression and odds ratio analysis are shown in Table 4.

## 4. Discussion

The present study evaluates the oncologic outcomes and clinical predictors of pathologic upstaging in a contemporary cohort of patients with MIBC treated with RARC, with a novel focus on stratification by NAC receipt status. Our findings suggest that upstaging portends significantly worse survival outcomes, particularly in patients who did not receive NAC. This underscores that the protective effect of NAC is only partial and does not fully mitigate the adverse disease biology in patients with MIBC. Importantly, our findings suggest that patients who were upstaged after NAC may represent non-responders to systemic therapy. These findings reinforce the importance of individualized risk assessment and offer new insight into how preoperative factors such as hydronephrosis, renal function, and clinical T stage may inform treatment selection.

Mitra et al. evaluated patients with MIBC (cT2N0M0) to identify the strongest predictors of pathologic upstaging following radical cystectomy [17]. Their findings showed that preoperative hydronephrosis, tumor growth pattern, depth of invasion, and age were significantly associated with upstaging, with preoperative hydronephrosis being the primary discriminator for risk classification based on a cross-validated model [18,19]. Our study echoes these findings by showing that among patients who were upstaged, preoperative hydronephrosis, higher clinical T stage, and lack of NAC are independent predictors for worse oncologic outcomes. The RETAIN-1 and RETAIN-2 trials employ a risk-adapted approach to identify patients who could be candidates for cystectomy-sparing active surveillance following NAC or neoadjuvant chemoimmunotherapy, respectively [20,21]. In this context, incorporating clinical and pathological elements that may help predict disease upstaging into the decision-making process could help identify patients who are better candidates for bladder-sparing avenues. Moreover, the persistence of clinicopathological predictors on repeat TURBT after NAC could help inform therapeutic algorithms and underscores the importance of preoperative risk identification for optimal treatment selection.

NAC works by eliminating micrometastatic disease and reducing the likelihood of distant recurrence [22]. Features like LVI or CIS in surgical specimens from patients who have already received NAC may indicate incomplete micrometastatic control, suggesting a higher residual disease burden than when these features are detected on initial TURBT and absent on surgical specimens. While our study did not identify CIS or LVI on TURBT as independent predictors of oncologic outcomes, our findings highlight the role of clinical and preoperative pathological factors in stratifying patients at risk of upstaging. Identifying high-risk individuals may help guide perioperative treatment selection, particularly in considering an intensified regimen such as combined chemo-immunotherapy, which has already shown improved disease control compared to chemotherapy alone [8].

Preoperative hydronephrosis was associated with more than a two-fold increase in the odds of disease upstaging, reinforcing its role as a marker of aggressive tumor biology. This association is likely explained by tumor invasion of the ureteral orifice, leading to urinary outflow obstruction, higher local tumor burden, and an increased probability of extravesical extension. This finding aligns with studies reporting worse outcomes among patients with bladder cancer and hydronephrosis, particularly when bilateral [23,24]. Unilateral and bilateral hydronephrosis in patients with bladder cancer is observed in approximately 13% and 3% of patients, respectively [23]. Our study builds upon this by demonstrating that hydronephrosis remains an independent predictor of both disease upstaging and survival in a contemporary cohort of robotic-assisted cystectomies, and that its adverse prognostic impact persists regardless of NAC receipt. Importantly, these findings highlight the ongoing relevance of hydronephrosis in patient selection and risk stratification for multimodal strategies, including bladder-sparing protocols, NAC, and emerging chemo-immunotherapy approaches. Beyond portending a worse prognosis, its presence frequently precludes enrollment in bladder-sparing trials. Zlotta et al. compared trimodal therapy with radical cystectomy in 722 patients with MIBC. Only patients without bilateral hydronephrosis and impaired renal function were eligible for trimodal therapy. Even among those with unilateral hydronephrosis, the 5-year metastasis-free survival was inferior compared to patients without hydronephrosis (*p* = 0.051). The unequivocal finding that hydronephrosis confers worse outcomes should be interpreted in the context of patients not being eligible to undergo bladder-sparing treatment options. Of 282 patients who were treated with trimodal therapy in this study, salvage cystectomy was performed in a relatively low number (n = 38, 13%), the vast majority due to an invasive recurrence [13]. Hydronephrosis may reflect a more advanced disease state and deprive patients of the opportunity for bladder preservation avenues. Given the inherent detriments of hydronephrosis and impaired renal function in many patients with bladder cancer, designing bladder-sparing clinical trials for this population is of utmost importance. The BC2001 trial evaluated the efficacy of fluorouracil and mitomycin C combined with radiotherapy versus radiotherapy alone in patients with MIBC. Locoregional disease control for the combination arm was 67% (95% CI, 59–74) vs. 54% (95% CI, 46–62) with RT alone, and the 5-year overall survival for the combination arm was 48% (95% CI, 40–55) vs. 35% (95% CI, 28–43) for RT alone [25]. However, no efforts to evaluate the efficacy of this intervention in patients with hydronephrosis were recorded. The RTOG 0712 study compared the effectiveness of fluorouracil with cisplatin and a twice-daily radiation regimen against low-dose gemcitabine and once-daily radiation as part of bladder preservation regimens. However, no patient with tumor-related hydronephrosis was eligible for inclusion [26]. In the context of multi-agent bladder-sparing trials, the detriment of hydronephrosis goes beyond oncologic outcomes, as it limits the accrual ability of this population to clinical trials.

Accurate clinical staging for bladder cancer remains a challenge, as conventional imaging techniques (CT and MRI) can underestimate disease burden. While cross-sectional imaging with CT and MRI scans remains the most widely used modality, its sensitivity for staging bladder cancer is limited (50–85% for CT and 75–85% for MRI) [27,28]. For this reason, adopting emerging strategies that incorporate liquid biopsy may provide additional tools to detect locally advanced disease and predict relapse in patients undergoing radical cystectomy. Sfakianos et al. demonstrated that detectable circulating tumor DNA (ctDNA) after radical cystectomy was independently associated with shorter disease-free survival. Indeed, a positive ctDNA signature at pre-established time points (before cystectomy, during the molecular residual disease window, and during the surveillance window) was the only risk factor independently associated with shorter disease-free survival [29]. In another study, longitudinal ctDNA assessment identified metastatic relapse with a sensitivity of 94% and specificity of 98% in NAC-treated patients. Additionally, pre- and postoperative ctDNA positivity also predicted worse outcomes in NAC-naïve patients [30]. These findings exemplify the growing potential of ctDNA to refine prognostication and monitoring treatment response in patients with MIBC. In this context, ctDNA dynamics could serve as a complementary tool to traditional clinical predictors, such as the ones highlighted in our paper, potentially improving risk stratification and guiding patient selection for multimodal strategies.

The results of this study should be interpreted in the context of the limitations inherent to retrospective studies. Our data only includes patients with MIBC who underwent RARC, so all data of patients who initiated NAC but were not surgical candidates were excluded, leading to a potential selection bias of patients with more aggressive disease. Bimanual examination and the use of CT or MRI during clinical staging were not captured, which may increase the heterogeneity of our findings. Because clinical and radiological characteristics were recorded at diagnosis, we were unable to ascertain whether hydronephrosis was chronic prior to bladder cancer diagnosis, which could limit the interpretation of its etiology in some patients. We did not explore the role of prior pelvic radiotherapy in disease upstaging, which may limit the interpretability of our results for this population. The potential influence of surgical approach (robotic versus open) on outcomes was not assessed, as comparison of surgical techniques was beyond the scope of this analysis. Additionally, all data were collected from patients treated at a single tertiary-level care center, which decreases the generalizability of results. Nonetheless, the present study benefits from including various ages, ethnicities, and tumor histologies.

## 5. Conclusions

Despite advancements in diagnostic and therapeutic strategies, the discrepancy between clinical and pathological staging in MIBC remains a significant challenge, as current assessment tools often underestimate disease burden. While NAC demonstrated a protective effect, it did not eliminate the risk of upstaging, emphasizing the need for improved risk stratification. Preoperative hydronephrosis was the strongest predictor of upstaging, highlighting its potential role in refining patient selection for multimodal treatment approaches. Given the negative impact of upstaging on oncologic outcomes, clinical vigilance is warranted when encountering patients with high-risk features. Future efforts should focus on integrating advanced imaging modalities, molecular biomarkers, and risk-adaptive treatment strategies to enhance staging accuracy and optimize therapeutic decision-making in patients with MIBC.

## Figures and Tables

**Figure 1 cancers-17-03265-f001:**
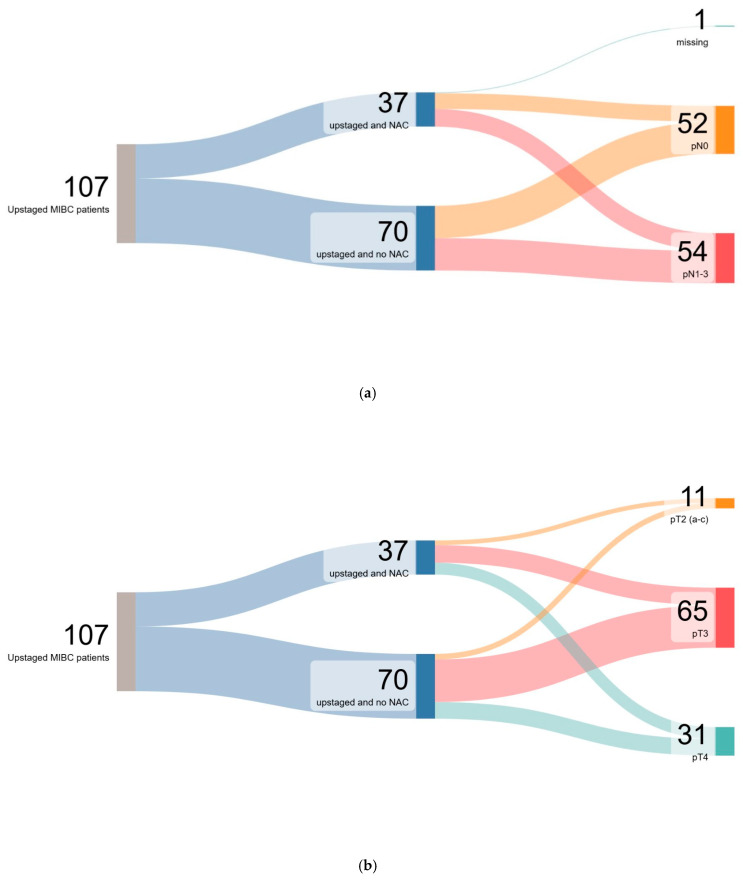
Sankey diagram showing upstaging status based on pathologic node (**a**) and tumor stage (**b**).

**Figure 2 cancers-17-03265-f002:**
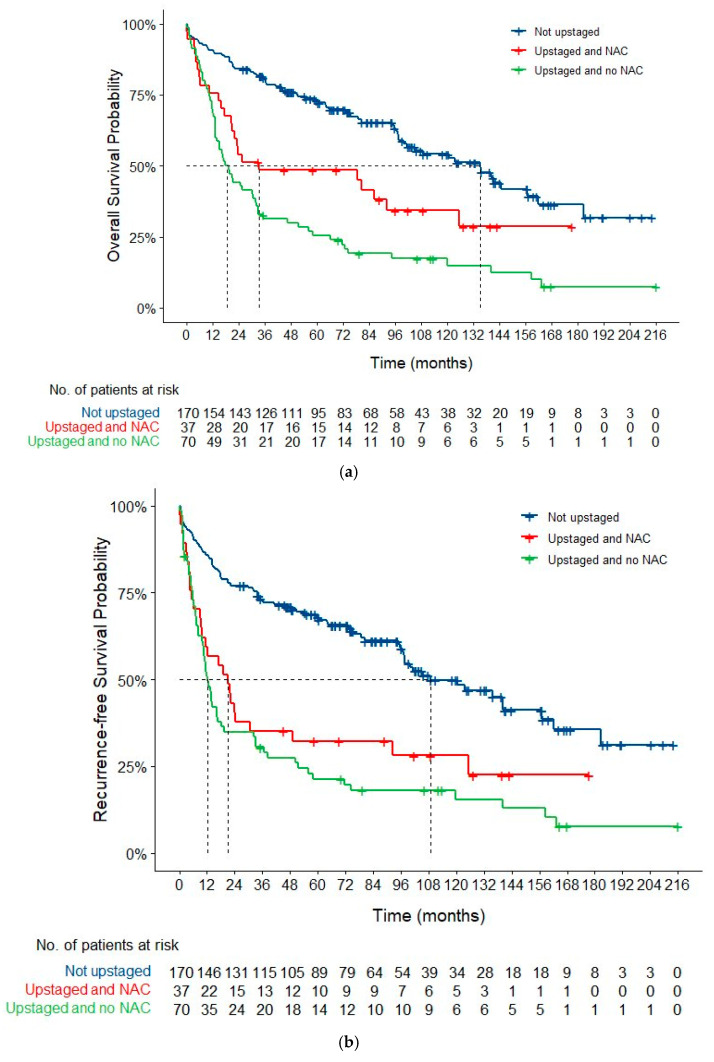
Overall survival (**a**) and recurrence-free survival (**b**) for patients who were not upstaged, upstaged with prior NAC, and upstaged without prior NAC.

**Table 1 cancers-17-03265-t001:** Baseline patient clinical and demographic characteristics.

	Overall (N = 277)	Not Upstaged (N = 170)	Upstaged and NAC (N = 37)	Upstaged and no NAC (N = 70)	*p*-Value
**Age at surgery**					**<0.001**
Median (IQR)	72.0 [65.0, 79.0]	71.0 [64.0, 77.0]	69.0 [60.0, 75.0]	77.0 [68.0, 82.8]	
**Sex, No. (%)**					0.12
Female	59 (21%)	31 (18%)	7 (19%)	21 (30%)	
Male	218 (79%)	139 (82%)	30 (81%)	49 (70%)	
**Race/ethnicity, No. (%)**					0.09
Hispanic	27 (10%)	18 (10%)	4 (11%)	5 (7%)	
Non-Hispanic black	5 (2%)	1 (1%)	1 (3%)	3 (4%)	
Non-Hispanic white	200 (72%)	129 (76%)	22 (59%)	49 (70%)	
Others	45 (16%)	22 (13%)	10 (27%)	13 (19%)	
**BMI**					0.08
Median (IQR)	26.8 [24.1, 30.9]	27.1 [24.6, 31.3]	27.9 [22.4, 30.5]	25.8 [23.1, 28.7]	
**ASA Score, No. (%)**					0.66
2	25 (9%)	17 (10%)	2 (6%)	6 (9%)	
3	174 (63%)	107 (63%)	26 (70%)	41 (59%)	
4	78 (28%)	46 (27%)	9 (24%)	23 (32%)	
**Tumor histology, No. (%)**					0.09
Pure urothelial	239 (86%)	150 (88%)	34 (92%)	55 (79%)	
Mixed histology or variant	38 (14%)	20 (12%)	3 (8%)	15 (21%)	
**Preoperative hydronephrosis, No. (%)**					**0.008**
No	197 (71%)	132 (78%)	21 (57%)	44 (63%)	
Yes	80 (29%)	38 (22%)	16 (43%)	26 (37%)	
**Pathologic node stage, No. (%)**					**<0.001**
N0	219 (79%)	167 (98%)	17 (46%)	35 (50%)	
N1–3	54 (20%)	0 (0%)	19 (51%)	35 (50%)	
Missing	4 (1%)	3 (2%)	1 (3%)	0 (0%)	
**Clinical T stage, No. (%)**					0.15
T2	234 (85%)	140 (82%)	29 (78%)	65 (93%)	
T3	25 (9%)	18 (11%)	5 (14%)	2 (3%)	
T4	18 (6%)	12 (7%)	3 (8%)	3 (4%)	
**Pathologic T stage, No. (%)**					
pT0	51 (18%)	51 (30%)	0 (0%)	0 (0%)	
pT1	14 (5%)	14 (8%)	0 (0%)	0 (0%)	
pT2	8 (3%)	7 (4%)	1 (3%)	0 (0%)	
pT2A	33 (12%)	30 (18%)	2 (5%)	1 (1%)	
pT2B	33 (12%)	26 (15%)	2 (5%)	5 (7%)	
pT2C	3 (1%)	3 (2%)	0 (0%)	0 (0%)	
pT3	6 (2%)	0 (0%)	1 (3%)	5 (7%)	
pT3A	46 (17%)	3 (2%)	14 (38%)	29 (41%)	
pT3B	18 (6%)	2 (1%)	4 (11%)	12 (18%)	
pT4a	1 (1%)	0 (0%)	0 (0%)	1 (1%)	
pT4A	30 (11%)	4 (2%)	13 (35%)	13 (19%)	
pT4B	4 (1%)	0 (0%)	0 (0%)	4 (6%)	
pTA	10 (4%)	10 (6%)	0 (0%)	0 (0%)	
pTIS	20 (7%)	20 (12%)	0 (0%)	0 (0%)	
**Concomitant CIS on TURBT, No. (%)**					0.52
No	238 (86%)	144 (85%)	31 (84%)	63 (90%)	
Yes	39 (14%)	26 (15%)	6 (16%)	7 (10%)	
**LVI on TURBT, No. (%)**					0.39
No	255 (92%)	159 (93%)	34 (92%)	62 (89%)	
Yes	22 (8%)	11 (7%)	3 (8%)	8 (11%)	
**Preoperative renal function**					**0.02**
Median (IQR)	1.08 [0.90, 1.28]	1.05 [0.87, 1.26]	1.07 [0.93, 1.26]	1.17 [0.99, 1.45]	

ASA, American Society of Anesthesiology; BMI, body mass index; CIS, carcinoma in situ; LVI, lymphovascular invasion; NAC, neoadjuvant chemotherapy; TURBT, transurethral resection of bladder tumor.

**Table 2 cancers-17-03265-t002:** Multivariable analysis for overall survival.

		No. (%)	Hazard Ratio (95% CI, *p*-Value)
**Treatment group, No. (%)**	Not upstaged	170 (61%)	Reference
	Upstaged and NAC	37 (14%)	1.99 (1.23–3.22, ***p* = 0.005**)
	Upstaged and no NAC	70 (25%)	3.18 (2.21–4.55, ***p* < 0.001**)
**Age at surgery**	Mean ± SD	70.9 ± 10.5	1.03 (1.01–1.05, ***p* = 0.004**)
**BMI**	Mean ± SD	27.7 ± 5.5	0.97 (0.94–1.00, *p* = 0.09)
**ASA Score**	Mean ± SD	3.2 ± 0.6	1.23 (0.93–1.63, *p* = 0.15)
**Preoperative hydronephrosis, No. (%)**	No	197 (71%)	reference
	Yes	80 (29%)	1.44 (1.02–2.04, ***p* = 0.03**)
**Clinical T stage, No. (%)**	T2	234 (84%)	reference
	T3	25 (9%)	1.47 (0.85–2.55, *p* = 0.16)
	T4	18 (7%)	3.18 (1.79–5.66, ***p* < 0.001**)
**Preoperative renal function**	Mean ± SD	1.1 ± 0.4	1.31 (0.93–1.86, *p* = 0.12)

ASA, American Society of Anesthesiology; BMI, body mass index; NAC, neoadjuvant chemotherapy.

**Table 3 cancers-17-03265-t003:** Multivariable analysis for recurrence-free survival.

		No. (%)	Hazard Ratio (95% CI, *p*-Value)
**Treatment group, No. (%)**	Not upstaged	170 (61%)	Reference
	Upstaged and NAC	37 (14%)	2.49 (1.58–3.94, ***p* < 0.001**)
	Upstaged and no NAC	70 (25%)	3.02 (2.11–4.31, ***p* < 0.001**)
**Age at surgery**	Mean ± SD	70.9 ± 10.5	1.03 (1.01–1.04, ***p* = 0.003**)
**BMI**	Mean ± SD	27.7 ± 5.5	0.98 (0.95–1.01, *p* = 0.14)
**Preoperative hydronephrosis, No. (%)**	No	197 (71%)	Reference
	Yes	80 (29%)	1.43 (1.02–2.01, ***p* = 0.04**)
**Clinical T stage, No. (%)**	T2	234 (84%)	reference
	T3	25 (9%)	1.95 (1.18–3.23, ***p* = 0.009**)
	T4	18 (7%)	3.77 (2.19–6.47, ***p* < 0.001**)
**Preoperative renal function**	Mean ± SD	1.1 ± 0.4	1.45 (1.05–2.00, ***p* = 0.02**)

BMI, body mass index; NAC, neoadjuvant chemotherapy.

**Table 4 cancers-17-03265-t004:** Logistic regression and multivariable analysis of clinicopathologic variables predicting upstaging status adjusting for covariates shown in Table 2 and Table 3.

		Not Upstaging (N = 170)	Upstaging (N = 107)	Odds Ratio (95% CI, *p*-Value)
Age at surgery	Mean ± SD	70.0 ± 10.5	72.2 ± 10.3	1.02 (1.00–1.05, ***p* = 0.09**)
Preoperative hydronephrosis, No. (%)	No	132 (78%)	65 (61%)	reference
	Yes	38 (22%)	42 (39%)	2.24 (1.31–3.81, ***p* = 0.003**)

## Data Availability

The datasets generated and/or analyzed during the current study are not publicly available, but are available from the corresponding author on reasonable request.

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
