# Peer review of "Clinical Predictors and Prognostic Significance of Pathologic Disease Upstaging at Radical Cystectomy in Patients with Muscle-Invasive Bladder Cancer"

_cancers, 2025, doi:10.3390/cancers17193265_

Round 1
Reviewer 1 Report
Comments and Suggestions for Authors
The authors aimed to evaluate clinical and pathological predictors of pathologic upstaging (pUS) stratifying by neoadjuvant chemotherapy (NAC) receipt, among patients undergoing robot-assisted radical cystectomy (RARC) for bladder cancer (BCA). They relied on retrospective study which enrolled 277 patients over 16 years (2004-2020). They found that preoperative hydronephrosis showed higher odds of upstaging [OR 2.24 (95% CI, 1.31–3.81), p=0.003].
The authors defined pUS as an increase in either T or N stage per surgical pathology report. However, the pUS seemed just to be a surrogate of NAC response. Intuitively, those who upstaged (or did not response to NAC if administeres) exhibited worse OS and RFS. RFS should be defined. However, we can interpret these results as "patients who upstaged may be likely no NAC responders". This should be specifically addressed. How many patients responded to NAC? Which were the criteria for NAC response?
The role of hydronephrosis in pUS should be better defined. Which is the biological rational?
Moreover, what was the hydronephrosis cause? A concomitant UTUC (data should be provided) or an obstructive BCA tumour? How many of these patients were chronically hydronephrotic?
Were these patients exposed to previous radiotherapy? (PMID 40586948)
Any influence of surgical approach ? (PMID 40152714)
Author Response
Reviewer 1: The authors aimed to evaluate clinical and pathological predictors of pathologic upstaging (pUS) stratifying by neoadjuvant chemotherapy (NAC) receipt, among patients undergoing robot-assisted radical cystectomy (RARC) for bladder cancer (BCA). They relied on retrospective study which enrolled 277 patients over 16 years (2004-2020). They found that preoperative hydronephrosis showed higher odds of upstaging [OR 2.24 (95% CI, 1.31–3.81), p=0.003].
Reviewer 1: The authors defined pUS as an increase in either T or N stage per surgical pathology report. However, the pUS seemed just to be a surrogate of NAC response. Intuitively, those who upstaged (or did not response to NAC if administered) exhibited worse OS and RFS. RFS should be defined.
- Response to reviewer: We thank the reviewer for this helpful suggestion. We have now clarified the definition of recurrence-free survival (RFS) in the Methods section (2.2 Statistical Analysis), specifying that RFS was defined as the time from surgery to local or distant disease recurrence based on histological or radiological evidence. This can be found in lines 109-111.
Reviewer 1: However, we can interpret these results as "patients who upstaged may be likely no NAC responders". This should be specifically addressed. How many patients responded to NAC? Which were the criteria for NAC response?
- Response to reviewer: We thank the reviewer for this insightful comment. While we agree that patient who were upstaged may be likely non responders to NAC, we also found that patients who did not receive NAC experienced disease upstaging, which likely reflects the limitations of clinical staging tools and disease understaging at baseline. We have underscored this in the discussion (lines 207-209). While response to NAC was not assessed for this analysis, the rate of upstaging in patients who received NAC (47%) compared to patients who did not (62%) can serve as a surrogate for disease response to NAC (lines 128-130). The criteria for response was based on surgical pathology results.
Reviewer 1: The role of hydronephrosis in pUS should be better defined. Which is the biological rationale? Moreover, what was the hydronephrosis cause? A concomitant UTUC (data should be provided) or an obstructive BCA tumour? How many of these patients were chronically hydronephrotic?
- Response to reviewer: We thank the reviewer for these insightful comments. We have clarified in the Methods that patients with concomitant UTUC were excluded, and hydronephrosis unrelated to the bladder cancer diagnosis was not considered in this analysis (lines 89-91). In the Discussion, we expanded the biological rationale, emphasizing that hydronephrosis likely reflects tumor invasion of the ureteral orifice, urinary outflow obstruction, and higher local tumor burden, which explain its association with upstaging. Finally, in the Limitations, we acknowledge that we were unable to capture whether hydronephrosis was chronic prior to bladder cancer diagnosis. These changes are found in lines 240-242.
Reviewer 1: Were these patients exposed to previous radiotherapy? (PMID 40586948)
- Response to reviewer: We thank the reviewer for this important question and for providing this reference. Patients who had received prior pelvic radiotherapy were not included in our cohort. We have also expanded our limitations section to mention that the impact of prior radiotherapy on disease upstaging was not evaluated, and results may not be generalizable to this patient population. These changes are found in lines 299-306.
Reviewer 1: Any influence of surgical approach? (PMID 40152714)
- Response to reviewer: We thank the reviewer for this thoughtful point and for providing this reference. Our study was limited to patients undergoing robotic-assisted radical cystectomy. For this reason, a comparison between surgical techniques was beyond the scope of this analysis. We have clarified this in the limitations section. Changes are seen in lines 299-306
Reviewer 2 Report
Comments and Suggestions for Authors
The authors presented an interesting topic, clearly indicating the need for better qualification for neoadjuvant therapy prior to the radical surgery. Especially in the era of current immunotherapy it should be brought to light. The authors decided to evaluate retrospectively the clinical and pathological predictors of pathologic upstaging in a cohort of patients who received neoadjuvant chemotherapy and underwent robot-assisted radical cystectomy (RARC), stratifying groups based on neoadjuvant chemotherapy-disease status.(n=277, single centre). These make a strong bias (single center, retrospective), although mentioned in the limitations.
The authors conlcuded that preoperative hydronephrosis was the strongest predictor for pathologial upstaging, independent of other baseline covariates. Based on earlier papers, it is well known that 50% risk and 90 % risk, respectively, exist in case of uni – and bilateral hydronephrosis. Could you comment on that in which way your result is novel?
For 16 years, the authors enrolled < 300 cases, which in turn gives approx. 20 cases annually. Could you comment on that in terms of not fulfilling the definition of high volume center for the treatment of MIBC?
Please consider subparagrahphs in the section of results. On Page 3, lines 125-137 please define the groups you compare (e.g. Pathologic node stage differed significantly across the groups – Line 129, e.g. as you described in figure 1).
Author Response
Reviewer 2: The authors presented an interesting topic, clearly indicating the need for better qualification for neoadjuvant therapy prior to the radical surgery. Especially in the era of current immunotherapy it should be brought to light. The authors decided to evaluate retrospectively the clinical and pathological predictors of pathologic upstaging in a cohort of patients who received neoadjuvant chemotherapy and underwent robot-assisted radical cystectomy (RARC), stratifying groups based on neoadjuvant chemotherapy-disease status.(n=277, single centre). These make a strong bias (single center, retrospective), although mentioned in the limitations.
- Response to reviewer: We thank the reviewer for their insightful overview of our study.
Reviewer 2: The authors concluded that preoperative hydronephrosis was the strongest predictor for pathologial upstaging, independent of other baseline covariates. Based on earlier papers, it is well known that 50% risk and 90 % risk, respectively, exist in case of uni – and bilateral hydronephrosis. Could you comment on that in which way your result is novel?
- Response to reviewer: We appreciate the reviewer's insightful comment. While we acknowledge that prior studies have elucidated the prognostic implications of hydronephrosis in bladder cancer, we believe the novelty of our study lies in demonstrating that hydronephrosis remains an independent predictor of upstaging and worse survival outcomes in a contemporary cohort of robotic-assisted cystectomy cohort, and that its prognostic significance persists regardless of NAC receipt. We compare a specific patient population who suffered disease upstaging, and stratification based on NAC status underscores the role of clinical and pathological elements in predicting worse disease outcomes. We have specified this contrast with published literature in the Discussion section, and changes can be found in lines 245-251.
Reviewer 2: For 16 years, the authors enrolled < 300 cases, which in turn gives approx. 20 cases annually. Could you comment on that in terms of not fulfilling the definition of a high-volume center for the treatment of MIBC?
- Response to reviewer: We thank the reviewer for raising this point. The number of patients in our analysis is a small subset of our entire institutional database. We only included patients with T2-4N0M0BC, who had no history of RT or TMT, only patients with available upstaging data, who had sufficient data on EMR, and who had adequate follow up data available. Thus, the reported sample size does not reflect the overall cystectomy volume at our center.
Reviewer 2: Please consider subparagraphs in the section of results.
- Response to reviewer: We thank the reviewer for this helpful suggestion. We have revised the Results section to include subparagraphs with descriptive subheadings to improve clarity and readability.
Reviewer 2: On Page 3, lines 125-137, please define the groups you compare (e.g., Pathologic node stage differed significantly across the groups – Line 129, e.g., as you described in figure 1).
- Response to reviewer: We thank the reviewer for their comments. We have addressed this and specified in the results section the comparable frequency of pN1-3 patients observed between both upstaged cohorts (NAC and no NAC). This is reflected in lines 139-140.
Reviewer 3 Report
Comments and Suggestions for Authors
This is a well-written and timely manuscript that addresses an important clinical issue in muscle-invasive bladder cancer (MIBC): the predictors and implications of pathologic upstaging following radical cystectomy. The authors leverage a sizable single-institution cohort with long follow-up and robust statistical methodology. The finding that preoperative hydronephrosis is the strongest predictor of upstaging is both clinically relevant and consistent with prior literature, strengthening the validity of the results.
Introduction is clear an concise and creates a clear background for the paper.
Methods chapter is very well structured offering clear explanations for the study population and the statistical analysis. Statitical analysis is done in a very professional manner, showing great methodological rigor.
Results chapter is very clear and easy to understand even for the average reader. Figures and table are really well made and offer better insight and deepens the understanding of the subject.
Discussions are relevant for the subject and conclusions are concise.
For these reasons I consider this manuscript very good and I propose it for publishing.
Author Response
Reviewer 3: This is a well-written and timely manuscript that addresses an important clinical issue in muscle-invasive bladder cancer (MIBC): the predictors and implications of pathologic upstaging following radical cystectomy. The authors leverage a sizable single-institution cohort with long follow-up and robust statistical methodology. The finding that preoperative hydronephrosis is the strongest predictor of upstaging is both clinically relevant and consistent with prior literature, strengthening the validity of the results. Introduction is clear and concise and creates a clear background for the paper.
- Response to reviewer: We thank the reviewer for their thoughtful feedback.
Reviewer 3: Methods chapter is very well structured offering clear explanations for the study population and the statistical analysis. Statistical analysis is done in a very professional manner, showing great methodological rigor.
- Response to reviewer: We sincerely thank the reviewer for this positive feedback regarding the clarity of our Methods and the rigor of our statistical analysis. We appreciate this recognition.
Reviewer 3: Results chapter is very clear and easy to understand even for the average reader. Figures and table are really well made and offer better insight and deepens the understanding of the subject.
- Response to reviewer: We thank the reviewer for these kind comments regarding the clarity of our Results section and the quality of our figures and tables. We are pleased that the presentation effectively conveys our findings.
Reviewer 3: Discussions are relevant for the subject and conclusions are concise. For these reasons I consider this manuscript very good and I propose it for publishing.
- Response to reviewer: We sincerely thank the reviewer for this highly positive feedback and for recommending our manuscript for publication. We greatly appreciate the thoughtful review and support.
Reviewer 4 Report
Comments and Suggestions for Authors
The authors evaluated the pathologic upstaging at time of radical cystectomy at a single institution in a retrospective fashion. The association of pathologic upstaging with survival was evaluated. The authors noted that patients with hydronephrosis had a greater likelihood of upstaging at cystectomy. Specific comments and questions are listed below.
- Who completed the clinical staging? As this study is evaluating upstaging based on clinical staging, it is important to clearly state who, when and how the staging was performed. I would suggest that the clinical staging be completed by a radiologist, urologist and medical oncologist for agreement. There seems to be a very high percentage of clinical T2 patients in this series.
- Was all of the clinical staging completed in a similar manner? How many patients had clinical staging based on an MRI compared to a CT scan?
- Did all patients undergo a bimanual exam as part of clinical staging?
- When was the clinical staging completed in patients receiving NAC, before or after NAC? If after NAC, did any of these patients have resolution of hydronephrosis following NAC?
- Were all patients cisplatin eligible?
- The authors need to discuss the potential of ctDNA to better predict presence of locally advanced disease at time of cystectomy following NAC.
- The authors need to discuss the most accurate method of assessing stage prior to cystectomy and how it may be improved.
Author Response
Reviewer 4: The authors evaluated the pathologic upstaging at time of radical cystectomy at a single institution in a retrospective fashion. The association of pathologic upstaging with survival was evaluated. The authors noted that patients with hydronephrosis had a greater likelihood of upstaging at cystectomy. Specific comments and questions are listed below.
Reviewer 4: Who completed the clinical staging? As this study is evaluating upstaging based on clinical staging, it is important to clearly state who, when and how the staging was performed. I would suggest that the clinical staging be completed by a radiologist, urologist and medical oncologist for agreement. There seems to be a very high percentage of clinical T2 patients in this series.
- Response to reviewer: We thank the reviewer for this important comment. As specified in the methodology, clinical staging was determined at the time of diagnosis by the treating urologist, based on cystoscopy/TURBT findings and cross-sectional imaging (CT or MRI), and was documented in the electronic medical record. (Lines 84-86).
Reviewer 4: Was all of the clinical staging completed in a similar manner? How many patients had clinical staging based on an MRI compared to a CT scan?
- Response to reviewer: We thank the reviewer for this important comment. Unfortunately, the use of MRI or CT scans was not captured in our database. Institutional practice is to use CT scans, so we anticipate that a very small proportion of patients, if any, underwent MRI as part of their initial clinical staging. We have addressed this in our limitations sections (lines 299-306).
Reviewer 4: Did all patients undergo a bimanual exam as part of clinical staging?
- Response to reviewer: We thank the reviewer for this insightful question. This variable is not captured in our database. We have acknowledged it in our limitations section. (Lines 299-306)
Reviewer 4: When was the clinical staging completed in patients receiving NAC, before or after NAC? If after NAC, did any of these patients have resolution of hydronephrosis following NAC?
- Response to reviewer: We thank the reviewer for this helpful comment. Hydronephrosis status was recorded before NAC, and resolution following treatment was not assessed in this study.
Reviewer 4: Were all patients cisplatin eligible?
- Response to reviewer: We thank the reviewer for their inquiry. Not all patients were cisplatin eligible. Specifically, 45 patients received NAC and 118 patients did not receive NAC.
Reviewer 4: The authors need to discuss the potential of ctDNA to better predict presence of locally advanced disease at time of cystectomy following NAC.
- Response to reviewer: We thank the reviewer for this important suggestion. We have added a new paragraph in the Discussion addressing recent studies on ctDNA as a prognostic and predictive biomarker in MIBC. We highlight its ability to identify relapse and its potential to complement traditional clinical predictors such as hydronephrosis in refining risk stratification. These changes appear in Lines 279-294.
Reviewer 4: The authors need to discuss the most accurate method of assessing stage prior to cystectomy and how it may be improved.
- Response to reviewer: We thank the reviewer for this suggestion. We have revised the Discussion section to address the limitations of current cross-sectional imaging modalities (CT and MRI) while also highlighting the role of emerging approaches such as ctDNA assessment. We discuss how these novel strategies may complement traditional clinical predictors and improve staging accuracy, prognostication, and patient selection for optimal therapy. Lines 276-279
Reviewer 5 Report
Comments and Suggestions for Authors
This manuscript analyzed the predictive and prognostic factors for upstaging of muscle invasive bladder carcinoma (MIBC) at radical cystectomy. The authors retrospectively collected adequate patients and provided appropriate statistical analysis. The results revealed that hydronephrosis is the sole prognostic factor for upstaging and predicting less favorable clinical outcome.
The study is well designed and the manuscript is well structured and written. The results are clinically relevant and valuable.
Author Response
Reviewer 5: This manuscript analyzed the predictive and prognostic factors for upstaging of muscle invasive bladder carcinoma (MIBC) at radical cystectomy. The authors retrospectively collected adequate patients and provided appropriate statistical analysis. The results revealed that hydronephrosis is the sole prognostic factor for upstaging and predicting less favorable clinical outcome. The study is well designed and the manuscript is well structured and written. The results are clinically relevant and valuable.
- Response to reviewer: We thank the reviewer for this generous and positive assessment of our study and manuscript. We greatly appreciate the recognition of the study design, statistical approach, and the clinical relevance of our findings.
Round 2
Reviewer 1 Report
Comments and Suggestions for Authors
No further comments are needed.
Reviewer 4 Report
Comments and Suggestions for Authors
The authors have adequately addressed my comments